# Sharing Augmented Reality between a Patient and a Clinician for Assessment and Rehabilitation in Daily Living Activities †

Mariolino De Cecco [1,*], Alessandro Luchetti [1,*], Isidro Butaslac III [2], Francesco Pilla [3], Giovanni Maria Achille Guandalini [3], Jacopo Bonavita [3], Monica Mazzucato [4] and Kato Hirokazu [2]

1 Department of Industrial Engineering, University of Trento, 38123 Trento, Italy
2 Division of Information Science, Nara Institute of Science and Technology, Nara 630-0192, Japan
3 Rehabilitation Unit, Azienda Provinciale Servizi Sanitari, Pergine Valsugana, 38057 Trentino, Italy
4 Rare Diseases Coordinating Center, Rare Diseases Registry, Veneto Region, 38123 Padua, Italy
* Correspondence: mariolino.dececco@unitn.it (M.D.C.); alessandro.luchetti@unitn.it (A.L.)
† This article is a revised and expanded version of a paper entitled "Multidimensional Assessment of Daily Living Activities in a Shared Augmented Reality Environment. In Proceedings of the 2022 IEEE International Workshop on Metrology for Living Environment (MetroLivEn), Cosenza, Italy, 25–27 May 2022".

**Abstract:** In rehabilitation settings that exploit Mixed Reality, a clinician risks losing empathy with the patient by being immersed in different worlds, either real and/or virtual. While the patient perceives the rehabilitation stimuli in a mixed real–virtual world, the physician is only immersed in the real part. While in rehabilitation, this may cause the impossibility for the clinician to intervene, in skill assessment, this may cause difficulty in evaluation. To overcome the above limitation, we propose an innovative Augmented Reality (AR) framework for rehabilitation and skill assessment in clinical settings. Data acquired by a distributed sensor network are used to feed a "shared AR" environment so that both therapists and end-users can effectively operate/perceive it, taking into account the specific interface requirements for each user category: (1) for patients, simplicity, immersiveness, engagement and focus on the task; (2) for clinicians/therapists, contextualization and natural interaction with the whole set of data that is linked with the users' performances in real-time. This framework has a strong potential in Occupational Therapy (OT) but also in physical, psychological, and neurological rehabilitation. Hybrid real and virtual environments may be quickly developed and personalized to match end users' abilities and emotional and physiological states and evaluate nearly all relevant performances, thus augmenting the clinical eye of the therapist and the clinician-patient empathy. In this paper, we describe a practical exploitation of the proposed framework in OT: setting-up the table for eating. Both a therapist and a user wear Microsoft HoloLens 2. First, the therapist sets up the table with virtual furniture. Next, the user places the corresponding real objects (also in shape) to match them as closely as possible to the corresponding virtual ones. The therapist's view is augmented during the test with motion, balance, and physiological estimated cues. Once the training is completed, he automatically perceives deviations in the position and attitude of each object and the elapsed time. We used a camera-based localization algorithm achieving a level of accuracy of 5 mm with a confidence level of 95% for position and 1° for rotation. The framework was designed and tested in collaboration with clinical experts of Villa Rosa rehabilitation hospital in Pergine (Italy), involving both a set of patients and healthy users to demonstrate the effectiveness of the designed architecture and the significance of the analyzed parameters between healthy users and patients.

**Keywords:** metrology in living environments; activity of daily living; shared augmented reality scenario; camera calibration; object identification

## 1. Introduction

With the development of more high-performing and accessible novel technologies, the potential for their use in different applications is advancing continuously. Within such

technologies, Augmented Reality (AR) enables an extensive array of unique use cases, particularly for practical and real-world applications [1–3].

This research study examined an AR framework for the metrological evaluation of activities of daily living (ADLs) [4] that involves the participation of both the occupational therapist and patient as the end-user. Our use case focuses on rehabilitation, but it may be expanded to an immersive AR environment to assist physically challenged end-users at home with the assistance of their caregivers.

Brain injuries such as a stroke, traumatic brain injury, or brain tumor may have a slight to severe influence on a person's ability to perform ADLs independently. Factors to consider might include motor/sensory impairments, cognitive/perceptual insufficiencies, behavioral shortfalls, or visual discrepancies. The evaluation of the patient's abilities is based on a non-standardized method that measures the patient's performance using these markers: safety, efficiency, effort, and independence [5]. In the current workflow, occupational therapists give standardized examinations using manuals such as the Assessment of Motor and Process Skills (AMPS) [6], which makes them more trustworthy and consistent, but still has some remaining flaws. Moreover, in order to gain the ability to conduct the proper AMPS evaluation, continued training is necessary. Consequently, an assessment utilizing these modalities is impacted by the clinician's expertise and is therefore susceptible to mistakes and misinterpretations for less experienced therapists.

To overcome this constraint, new technologies, and enhanced measuring techniques help objectively evaluate the effectiveness of treatment and training programs. The use of AR as assistive technology in clinical settings is widely discussed in literature also for ADLs support [7–9]. In most situations, the decision to employ AR technologies rather than other technologies in the mixed reality spectrum is based on the notion that generally, subjects perform better in AR than in Virtual Reality (VR) in terms of self-to-environment-related movements as hand-eye coordination in VR gives a much higher extraneous cognitive load [10]. Additionally, in our prototype, the patient must manipulate physical items while seeing virtual information in AR to ensure the perception of the surrounding physical environment and the weight of these said objects.

The fundamental novelty of the proposed framework lies in the enhancement and support of the clinical eye [11] in a shared real/virtual environment that enables the evaluation/support in AR contexts for future ADLs scenarios by increasing empathy between actors [12]. The proposed prototype increases the therapist's involvement with the patient with the ability to access their multidimensional data in AR. Furthermore, it helps improve the patient's engagement by allowing interaction with virtual augmented information and real tools/utensils throughout the ADL exercise. Using the suggested framework, we built a specialized program for occupational therapists that incorporates the AR system with a robust, reliable, and accurate computer vision-based technique to assure the high metrological quality of the evaluation.

This paper can be divided as follows:

- In the introduction, the issues about ADL assessment are discussed.
- In the section that follows, the current status and implementations of ADL assessment are examined.
- In the third section, the proposed ADL framework is explained in detail.
- In the fourth section, the specific ADL scenario for setting up the table is introduced.
- In the fifth section, the algorithm, and techniques used for detecting and identifying the objects of interest are presented.
- In the sixth section, the outcomes of the metric evaluation based on the acquired data are investigated.
- In the seventh section, the findings of the user study conducted, which compared healthy testers to patients, are reported.
- In the eighth section, the attributes and benefits of the offline interface provided to the therapists are shown.
- In the final section, the conclusions reached are elucidated.

## 2. Related Work

In the literature, there is already much research that has dealt with the assessment of ADLs. For example, Oyeleke et al. [13] performed a study where they described a "Situ-centric learning automaton recommendation system" for guiding a resident of a smart home through a series of ADLs chores. Their recommendation system employed a customized machine learning algorithm based on a self-operating model that breaks down the ADL into sequential tasks. The output of their system would then be able to understand the user's intentions and motivations and provide relevant recommendations.

Another study is of Ohiwa et al. [14], where they proposed a system that monitors and advises the elderly based on their everyday activities. A wireless sensor network was used in the suggested system. A sensor device was designed and installed in the residence of elderly people to collect data in their homes. They presented a strategy for identifying the activity patterns of the elderly and experimentally verified this using data acquired from their residence. Their experimental assessment demonstrated that the sensor-collected sound data may be used to identify the activity pattern, including waking and sleeping time. Because of the developed learning model from the observed sound data, it is also feasible to identify a variation in the activity pattern. Their findings indicate that it is feasible to determine the activity pattern of older individuals based on their action parameters while considering their privacy and the ambient data collected from their residences.

Moreover, Karakostas et al. [15] proposed the deployment of a sensor network with a three-layer design to aid individuals with mild cognitive impairments with their everyday tasks. Their sensor network comprises modules that facilitate the representations of each device and, as a result, generates a universally homogeneous user interface. ADLs are tracked and detected based on information about the surrounding environment and can be viewed easily on the interface mentioned above. They conducted four case studies and evaluations which expanded to the data tracked from sleep, camera, plug, and kitchen sensors over a 3-month intervention.

Much work has been performed regarding technologies supporting the ADL of the patient. However, the help of the therapists' clinical eye (i.e., their assessment of the patient) during the ADL task has not yet been fully explored. The system described in this work does not replace the traditional workflow of the therapist-patient interaction during the ADL but instead promotes and deepens this interaction [16]. This interaction is bridged by having the patient see virtual guides that support their understanding of the ADL task that the therapist describes. Moreover, on the therapist's side, they can see the invisible current conditions of the patient (i.e., body and feet posture, heart rate) and can better understand the situation and make correct decisions and guidance about the ADL execution.

Shared Virtual and/or Augmented Reality has focused on sharing the same real and/or virtual environments or elements to foster collaboration between two human agents with the same roles and goals. Examples can be found in driving cars [17], in industrial settings [18] and, even, in human-robot interaction [19]. In clinical rehabilitation settings, where a physician shares the same AR with a patient, the tasks and roles of the two actors are completely different. However, shared AR has not yet been fully exploited due to the inherent difference between the two viewpoints. For this reason, we propose an original framework that shares the same environment augmenting it with the proper elements that consider the two different viewpoints.

## 3. ADL Framework

The prototype has been developed in the laboratory of Measurement, Instrumentation and Robotics of the University of Trento and set up inside the home automation apartment AUSILIA (Assisted Unit for the Simulation of Independent Activities) [20] at the rehabilitation hospital Villa Rosa in Pergine Valsugana (Italy). Figure 1 shows the framework tools used during the ADL assessment.

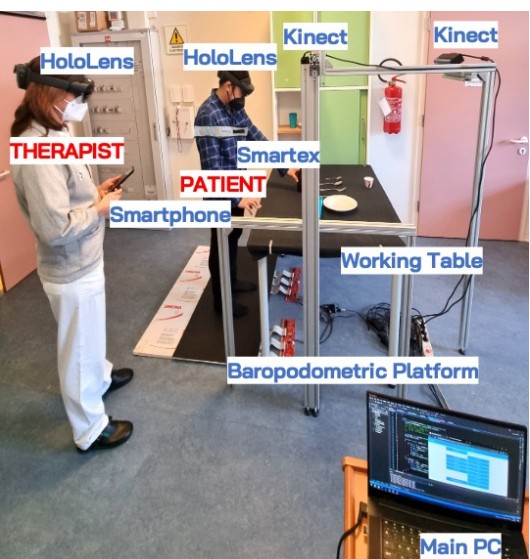

**Figure 1.** Framework setup in the AUSILIA apartment.

### 3.1. Visualization Devices

Both therapists and patients can see AR cues on Microsoft HoloLens 2 head-mounted displays. In addition, the therapist can manage and interact with information using a handheld device, such as a smartphone.

### 3.2. Distributed Measurement System

The measurement system consists of the following:

- Two time-of-flight depth cameras, such as the Microsoft Kinect v2, with the first one in front of the patient, used to determine where the position of the body joints are in 3D space [21]; the second one, above the table, used to capture an RGB image ($1920 \times 1080$) for the computer vision based-algorithm and to measure the height from the table and and check its orientation during the initial setup phase.
- A wearable band system developed by the company Smartex s.r.l of Navacchio (PI), Italy. It continuously monitors several physiological parameters. In particular, the system can simultaneously acquire the patient's electrocardiographic (ECG) and respiratory signals.
- The baropodometric platform used for non-invasive static and dynamic pressure measurement and body stability analysis is a customized model of the FreeMed family manufactured by the italian company Sensor Medica of Guidonia Montecelio (RM). The platform, which measures $56 \times 120$ cm, consists of two units, the sum of which results in 6000 24k gold-coated resistive sensors with frequency acquisition up to 400 Hz.
- The main PC, where all raw sensor data are processed, stored, and sent.

### 3.3. Software Development and Communication Protocols

The control interface for handheld devices such as smartphones was developed with the Node-RED programming tool. This development tool is useful for real-time data management and elaboration for IoT distributed systems. Its advantages include: open-source, visual programming ("flow-based programming"); fast development; lightweight; efficient MQTT (Message Queuing Telemetry Transport) client-server protocol.

All devices, including HoloLens, a smartphone, and the main computer, are connected over the same LAN. The MQTT protocol, based on TCP/IP, thanks to its reliability and lightness, allows the communication of data involving logic control (i.e., interface buttons, switches, and other controls). On the other hand, standard UDP (User Datagram Protocol) broadcasts data that concerns a large and continuous stream of information (i.e., platform data, Kinect data). The data transmission pipeline is shown in Figure 2.

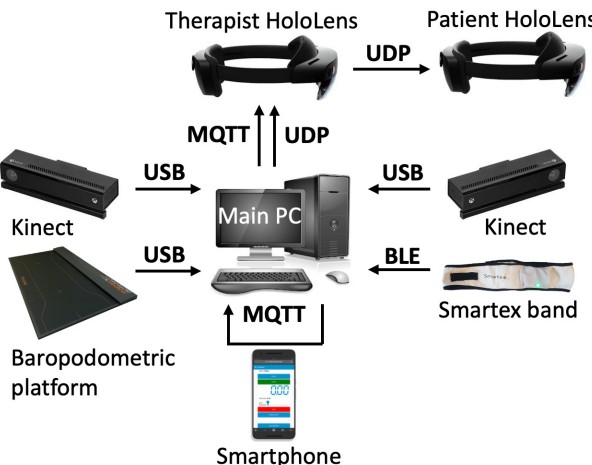

**Figure 2.** Data transmission pipeline.

The raw ECG and respiratory signal acquired by the Smartex band were processed and analyzed via Bluetooth Low Energy (BLE) in the main PC. HoloLens then received this data. In particular, for the analysis of the ECG and the respiratory signal, we took a 2 s time window to highlight any changes in physiological signals while the patient was performing short tasks. The Smartex band acquires the raw ECG signal with a frequency of 250 Hz. Data were successively filtered using a zero-phase passband Butterworth filter (cutoff frequencies, 0.1 Hz–20 Hz) and a modified version of the Pan–Tompkins algorithm was implemented to detect the R peaks [22]. The time differences between consecutive R peaks were calculated, obtaining the RR interval time series. For the patient's average heart rate, we considered the mean value of the punctual heart rate values within the 2 s time window. The breath rate was extracted from the raw respiratory signal (acquired at 50 Hz frequency) by removing the mean value and applying a zero-phase bandpass Butterworth filter (cutoff frequencies, 0.1 Hz–0.6 Hz). The peaks in the resulting signal were detected considering the following assumptions: a temporal distance greater than half of the average distance between all peaks and an amplitude greater than half of the average amplitude. As for the ECG signal, the considered breath rate was the average value within the 2 s time window.

Both Kinects are connected via USB to the main PC and are not used for synchronous acquisitions. The first is used during the table-setting task, and the second when the therapist presses the button to evaluate the error between virtual and real objects. We choose which sensor to use by enabling and disabling the USB port to which each Kinect is connected. For the first Kinect, we considered only the joints of the upper half of the body. The 3D coordinates of these joints are then converted from Kinect to a marker coordinate system using the transformation matrix calculated from the calibration. These data are then broadcasted via UDP to HoloLens at a rate of 30 Hz. The second Kinect acquires the RGB image to be provided as input to the computer vision-based algorithm, the output of which, in the form of deviations in the position and attitude of each object, is sent via UDP to HoloLens.

The FreeMed platform also connects to the main PC via USB. Data is collected using the C# program given by the FreeMed company, and broadcasted via UDP to HoloLens. The platform comprises 120 by 50 pressure sensors, with a total of 6000 small sensors. Each sensor returns a value between 0 and 255, with 0 being no pressure and 255 with max. This sensitivity is adjusted during the calibration phase.

Butterworth filters were applied to the Kinect and HoloLens data to reduce noise. A sixth-order Butterworth filter with 3 Hz cutoff frequency was selected for filtering both devices. Figure 3 summarizes the data processing flow chart of the main devices.

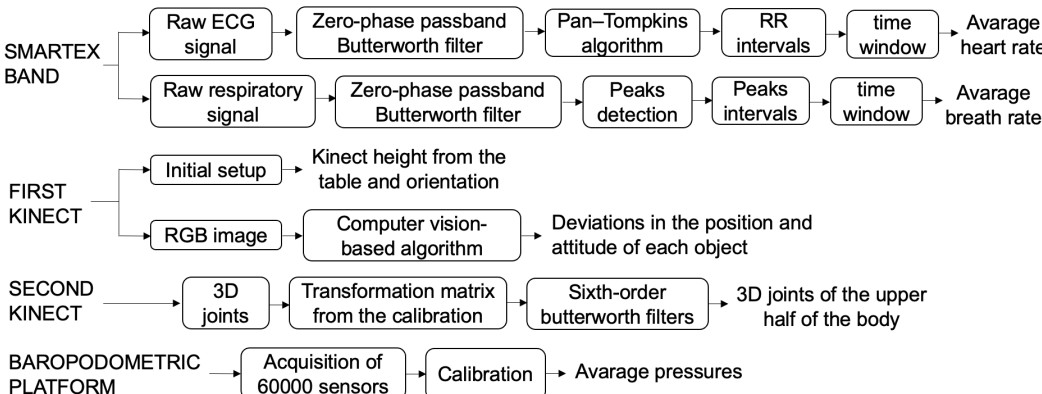

**Figure 3.** Data processing flow chart.

### 3.4. Extrinsic Parameters Calibration

For Kinect and the therapist's HoloLens to track the patient's kinematics in the same reference system, calibration is required (Figure 4). During the set-up phase, a marker with enough detectable feature points was used to derive a transformation matrix from Kinect camera coordinates to marker coordinates. The calibration process is repeated until an acceptable reprojection error is achieved. Vuforia SDK handles all the image target tracking for HoloLens. A spatial anchor is saved in the HoloLens using the same marker used for Kinect calibration. In this way, the Kinect and HoloLens can operate in the same reference system. Once calibrated, the marker can be removed at any time. Additional spatial anchors are saved in the therapist's HoloLens to define the reference systems of the working table and baropodometric platform. On the patient's HoloLens, however, only the spatial anchor related to the working table reference system is saved to operate in the same reference system as the therapist's HoloLens.

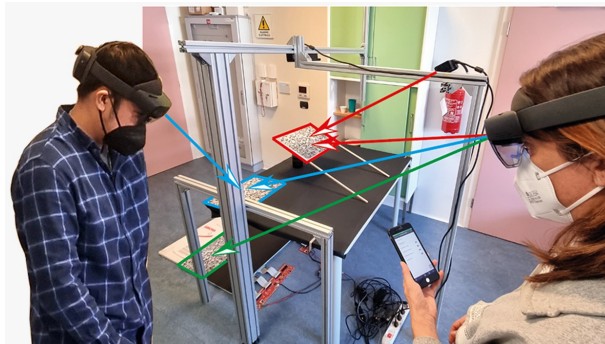

**Figure 4.** Spatial Anchors setting: the red image target is used by the therapist's HoloLens and the Kinect to operate in the same reference system; the blue target is used by the therapist's and the patient's Hololens to have the same reference system of the working plane; the green target is used only by the therapist's HoloLens to localize the baropodometric platform in space.

## 4. Specific ADL in the Kitchen Environment

The therapist assessing the patient during the instrumental ADL of setting the table is aided by a shared scenario in AR that can enhance his clinical assessment in an immersive and engaging way for the patient. The evaluation process involves the following steps:

(1).    Wearing a head-mounted Microsoft HoloLens 2, the therapist sets the table with virtual objects, Figure 5a. A handheld device's graphical interface allows the therapist to select the type and number of objects. Depending on the type of patient being assessed, therapist can adjust the complexity of the setup as needed. During this phase, the patient wearing another HoloLens 2 can view the virtual environment setup from his point of view, Figure 5b.

(2).    Once finished, the patient can view the virtual environment previously set up by the therapist and must try to match the virtual objects with the real ones.

(3).    Once the table setting is completed, the patient is asked to move his hands away from the table to avoid hiding real objects from the camera's view. Then, by pressing a button on the smartphone, the therapist estimates how far the real objects are from the virtual ones based on the position and angle errors that appear in AR next to each object with numbers following the therapist's gaze in Figure 6a. Numbers are displayed in different colors (green-yellow-red) according to the tolerance and, therefore, the threshold of error acceptability set by the therapist. If the algorithm does not find a match between a real object and a virtual object because, for example, the patient forgot to add the corresponding real object above the table, the associated virtual object is completely colored red. This indicates that the patient made an error with this virtual object; it is then up to the therapist to assess what kind of error because the algorithm could not return an output.

(4).    Another panel in AR summarizes the average angles and the average distances between the barycenters of the virtual and real models with the total task execution time, shown in Figure 6b.

(5).    Therapists can decide with a smartphone whether to display additional information about the patient in AR during the exercise session, such as the reconstruction of the patient's kinematics and angles between the limbs, the load distribution of the legs, and his physiological parameters (Figure 7).

(6).    At the end of each session, the therapist can decide to save all captured data to a text file.

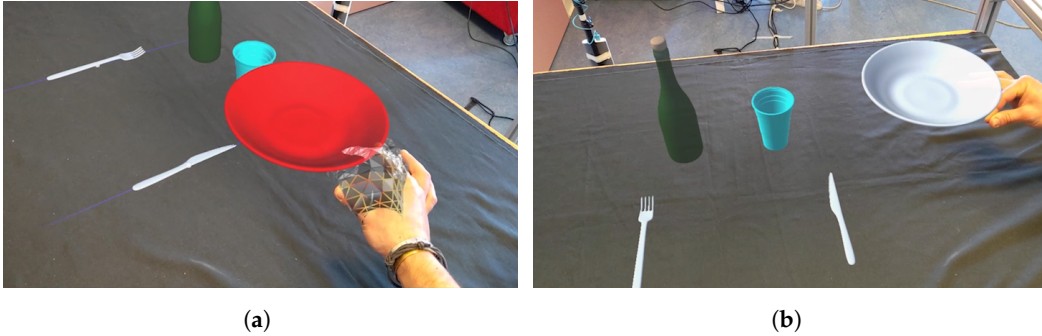

(**a**)                                                                                      (**b**)

**Figure 5.** Example of a shared AR environment from the field of view of the (**a**) therapist's HoloLens and (**b**) patient's HoloLens.

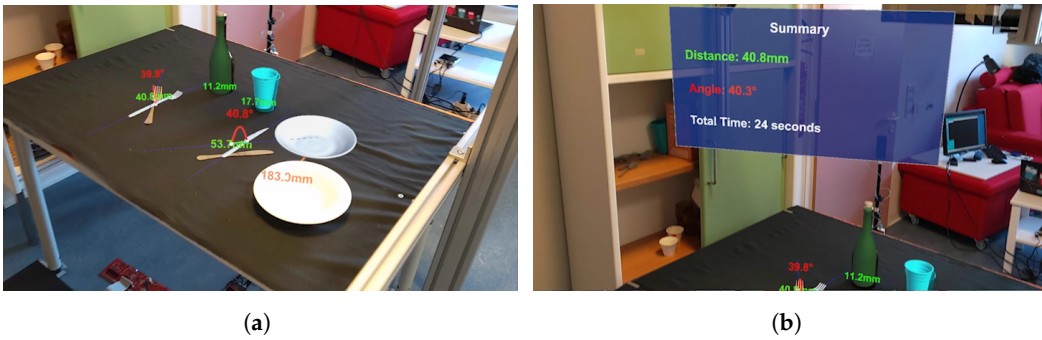

(**a**)                                                                                      (**b**)

**Figure 6.** (**a**) Example of errors visualization in AR via therapist's HoloLens 2 with (**b**) AR panel in which error averages and total time are summarized.

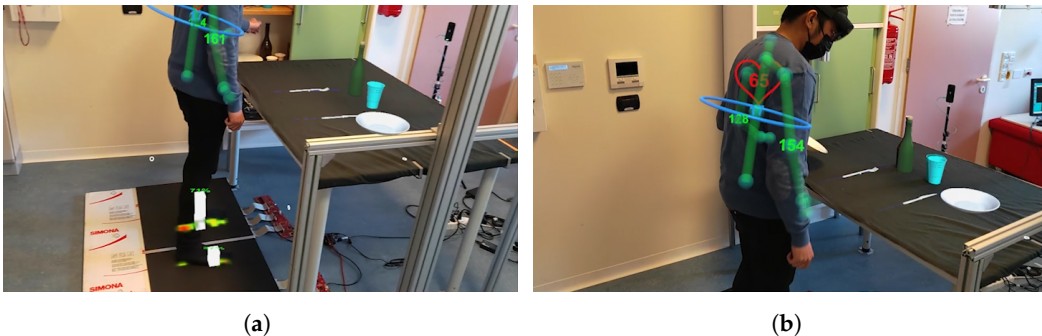

(**a**)　　　　　　　　　　　　　　　　(**b**)

**Figure 7.** Example of information in AR from the therapist's point of view on the (**a**) patient's lower and (**b**) upper body.

## 5. Algorithm for Object Segmentation, Localization & Identification

An algorithm was developed in a MATLAB environment to identify and locate real objects of interest placed on a table by a user. Following the processing of an RGB image, this algorithm can detect and identify such objects, as summarized in Figure 8.

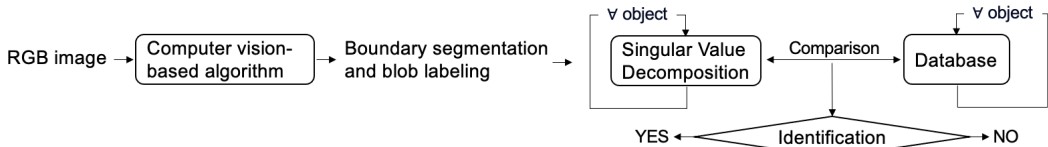

**Figure 8.** Flow chart of algorithm processing data.

It is not a real-time algorithm, but is executed only when a snapshot image is taken as input at the time the therapist presses the button to evaluate the error between virtual and real objects. In addition, the captured image only covers the plane of the table, so objects that are outside the camera's field of view are not considered in the object recognition process. Items to identify include polished stainless steel cutlery. A sandblasting process made them opaque and unaffected by the direction of light, overcoming the problem of reflections on their surface that could affect the result of the algorithm.

The algorithm can be divided into the following steps.

### 5.1. Segmentation and Localization

First, the RGB image captured by the Kinect fixed on top of the table was captured and processed in the following order:

(1).　Using a Kinect, grayscale images were acquired of the empty table and the same table covered with real objects.

(2).　Images were cropped to take into account only the table region of interest (ROI).

(3).　Each pixel was subtracted from the two previous images following background subtraction, and a threshold was selected to convert the result to a binary image.

(4).　The resulting mask was applied to the initial RGB image of the table set (Figure 9a), and a color-based threshold was applied to remove object shadows from the image (Figure 9b).

(5).　Next, flood-fill operations were performed on the hole pixels of the closed regions [23], as shown in Figure 9c.

(6).　A boundary label was applied to the filtered image [24].

(7).　Noise was removed by applying a threshold on the minimum number of pixels over the area of each labeled object.

(8).　The outer boundaries of each object were then traced [25], as shown in Figure 9d

(9).　Objects were localized by taking the mean of their boundary coordinates and by rotating them using Singular Value Decomposition (SVD).

(10).　In the end, a mask with each object-centered and aligned was stored.

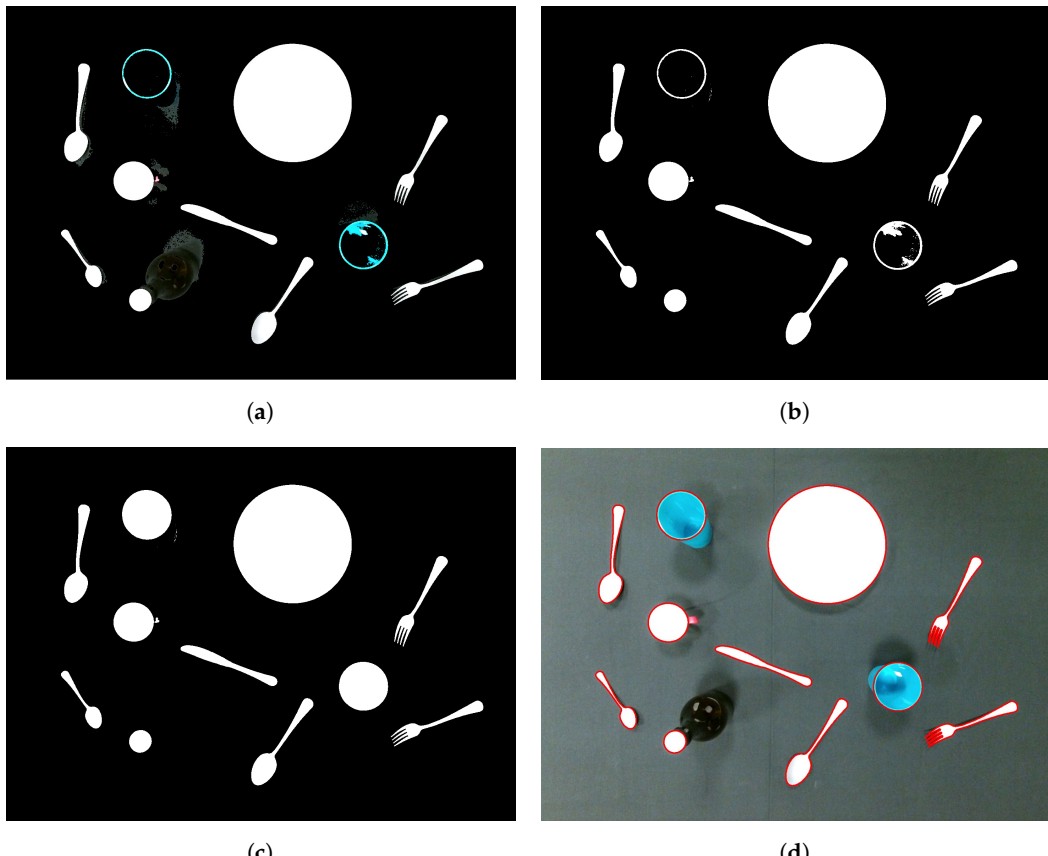

**Figure 9.** (**a**) Resulting of the mask applied to the original RGB image; (**b**) Color-based threshold to remove object shadows; (**c**) Flood-fill; (**d**) Boundary segmentation and blob labeling.

### 5.2. Identification

The previous image processing produces a binary image of each object segmented and aligned to the center of the initial image. Objects under consideration were compared to a previously created database using a cost function. The database was created using the same segmentation and realignment method as in the previous subsection, and the final labeling of the objects was carried out manually. Only one image for each object was required to initialise the database that will be referenced during matching ($REF_{IM}$).

Based on a set threshold, the input image ($IN_{IM}$) was compared to all objects in the database to identify the best match.

The first step is determining whether the areas between $IN_{IM}$ and $REF_{IM}$ are similar within 30%. If so, the cost function (CF) between them is calculated as follows:

$$CF = \frac{(1 - SC) + (1 - SA) + (1 - SSIM)}{3} \qquad (1)$$

where $SC$ is the score of similarity related to the object contours. In particular, the contour of $IN_{IM}$ is smoothed with a 2D Gaussian smoothing kernel with a standard deviation whose value changes according to the object's size. Then, the resulting image is converted to a binary image and multiplied by the contour of $REF_{IM}$ to check how many points of the two contours are in common. $SA$ is the score of similarity related to the object areas. It consists of the product of the two binary images of $IN_{IM}$ and $REF_{IM}$ to check how many points of the two areas are in common. $SSIM$ calculates the score related to the structural similarity between the $IN_{IM}$ and $REF_{IM}$. This score is a multiplicative combination of the three terms, namely the luminance, contrast, and structural term [26]. However, the black background is very predominant with respect to the size of the object when comparing $IN_{IM}$ and $REF_{IM}$ with $SSIM$. Therefore, both source images were cropped before comparison to make this score more sensitive to the objects in the images. The two new images have the same

size between them, i.e., 50% more than the dimensions of the largest object in $IN_{IM}$ and $REF_{IM}$, to be sure that the objects are still contained in the cropped images. All scores in Equation (1) are normalized. All terms are subtracted from the value 1 because we are looking for the minimum value of CF.

## 6. Metric Calibration of the Working Table

After initial camera calibration, metric analyses were performed to assess the implemented algorithm's performance in identifying real objects and estimating their position and orientation.

### 6.1. Camera Calibration

During camera calibration, the coordinates of each pixel on the CCD image sensor are compared with their real-world measurements. This is done by taking into account lens distortions, which are the most common monochromatic optical aberrations. At a fixed height of 80 cm, the Kinect camera captures an image of a planar pattern perpendicular to the table and in its center. The planar pattern consists of 55 Aruco markers located at the vertices of a grid with known positions. The geometrical centers and identifiers of the Aruco markers [27] were saved and compared to their locations in the environment.

An additional planar Aruco model (Figure 10a) was used to evaluate the calibration process and thus the accuracy of a random position on the table plane of dimensions $750 \times 1020$ mm. Once the set of random Aruco markers in the four corners was taken, the second time, the set of randomly placed Aruco markers in the center was taken, and the corresponding two-dimensional covariance matrices were computed. Covariance matrix results are shown in Figure 10b. As expected, the uncertainty ellipse around corners is larger due to the higher camera distortion.

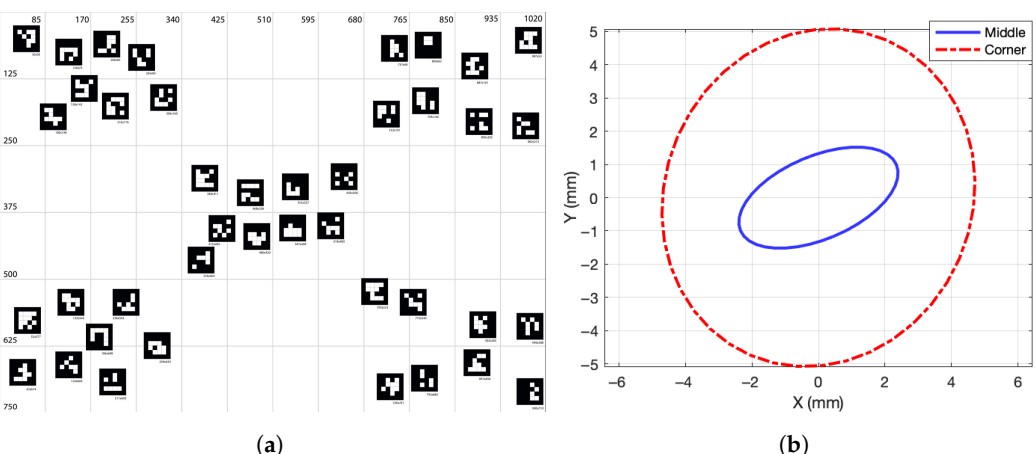

(a)          (b)

**Figure 10.** (**a**) Aruco markers plane for accuracy checking; (**b**) Ellipses of uncertainty in position (95% confidence level with $k = 2.4478$ [28]).

Moreover, the height of the objects used is different. For example, a bottle is much higher than cutlery which is flat on the table. Nevertheless, it has not been necessary to calibrate the camera at different heights because knowing the exact heights of the objects and the camera, with respect to the plane using trigonometric operations, we have always referred to the plane of the table.

### 6.2. Accuracy in the Image-Pattern-Recognition Tool

The reference system of the HoloLens 2 worn by the therapist and the user were initially set up by watching a square appear on a predefined pattern using the Vuforia Engine image-pattern-recognition tool and saving its position and orientation over time. It is possible that the two reference systems are not aligned with each other because image marker detection and rendering stability can be affected by several factors. The size of

the image marker and the resolution of the HMD camera do not affect the final accuracy because the HMD hardware and the image marker are the same for the therapist and the patient. What most affects the final accuracy is the distance and angle between the camera and the image marker. The authors in [29] provided $\leq 2°$ and $\leq 2\,$mm inclination angle and positional errors, respectively, in 70–75% of cases by using a holographic headset combined with an image-pattern-recognition tool. It could be a problem for the therapist's final assessment. On the other hand, except for a small error of different visualizations of the virtual objects in the shared virtual environment, it does not suffer of differences in position and attitude of each object between the real one and its virtual model. Everything is evaluated on board the patient's HMD with its reference system. Therefore, for our metric purpose of therapist assessment of patient exercise, the accuracy of the image pattern recognition tool is irrelevant between the therapist's HMD and the user's HMD.

### 6.3. Algorithm Accuracy for Object Segmentation, Localization & Identification

We performed rotation tests with a knife to evaluate the performance of the developed computer vision-based algorithm. In particular, Figure 11 shows a cropped RGB acquisition image of tests conducted using a manual rotation motion platform (Standa 126865) covered with black to facilitate the background subtraction and filtering process.

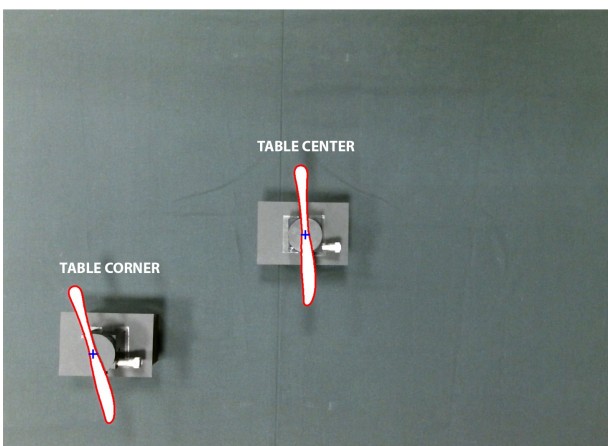

**Figure 11.** Cropped RGB image acquired for rotation tests in the two setups: in the center of the table and near a corner of the table.

Figure 11 shows both setups: one for tests conducted in the center of the table and one for tests near a corner.

For the rotation tests, 180 acquisitions were performed for each of the two setups from 0 to 360° with a step size of 2°. The decision to carry out these tests on both the center and sides of the acquired images was to estimate better the algorithm's performance over the entire table surface. The differences between the obtained rotations from the SVD algorithm and the one from the rotation motion platform taken as ground truth are shown as histograms of residuals for the two different setups in Figure 12a. The histogram spread for the setup at the center of the table is smaller than for the setting near the corner due to the higher camera distortion. However, the residual in estimating rotations for the localization algorithm in general over the entire table surface is less than 1°. During the same rotation tests in the two setups, the object center positions at each step were calculated as the mean of its boundary coordinates. The results of the object centers at each step are shown in Figure 12b.

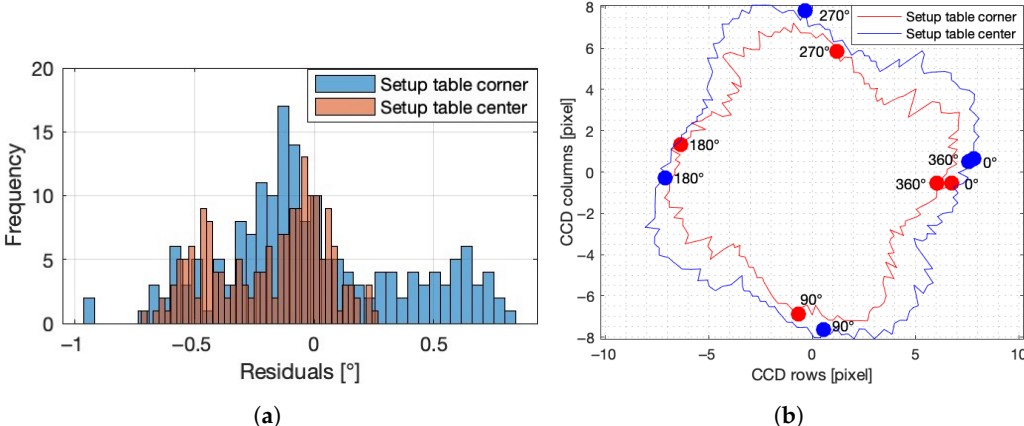

(a)                                                          (b)

**Figure 12.** (**a**) Histograms of residuals and (**b**) object center positions during rotations in the two setups.

## 7. User Study

An experimental test campaign approved by the ethics committee was also carried out with patients and healthy testers, as shown in Figure 13. This preliminary user study aims to assess the statistical significance of the data collected. The parameters analyzed are:

- errors in object placement;
- execution time;
- hand speed;
- breath rate;
- heart beat;
- pressure distribution.

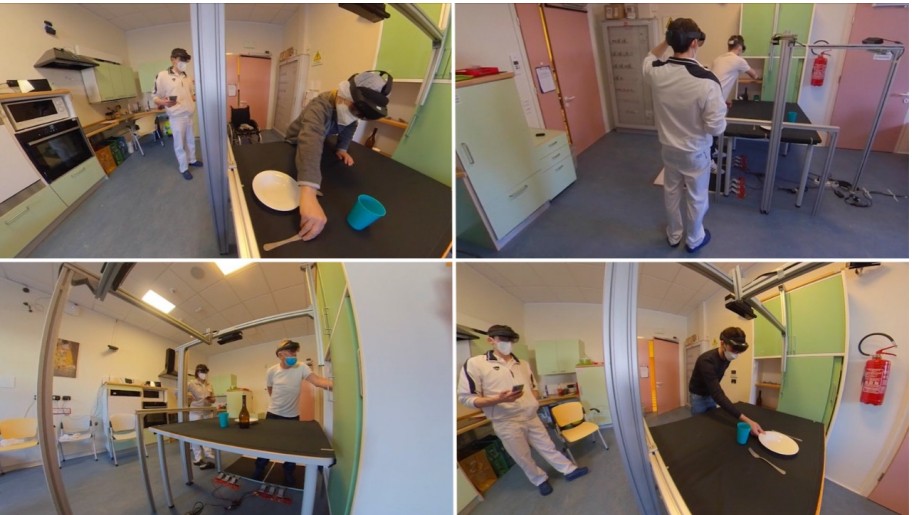

**Figure 13.** User study with four random testers among the eight participants.

In particular, errors in object placement refer to the median error in position and angle between real and virtual objects above the table; execution time quantifies the time between the moment the tester starts to pick up the first real object and the moment he finishes arranging all the objects on the table; hand speed is obtained from the acquired kinematics of the tester. For statistical analysis, we consider its maximum value. When a tester had a problem in either joint, he was forced to perform the entire test using only that one; physiological parameters, such as breath rate and heart beat, are calculated with respect to variations from their basal values; *pressure distribution* of each foot is analyzed with Warren Sarle's bimodality coefficient (BC) [30]. BC lies within a range of 0 to 1, where values greater than 0.555 indicate bimodal or multimodal data distributions.

Eight subjects participated in the tests voluntarily after signing a consent form. These were divided into two groups: three were patients, and five were healthy users. The selected patients, with ages between 19 and 69 years, including one female, have different pathologies:

- User 1, C5 incomplete tetraplegia, the major deficit in the left hand.
- User 2, cerebellar ataxia, balance, and stability problems.
- User 3, tetraparesis from Guillain–Barré outcomes, upper limb manipulation deficit.

None of them reported having experience with AR technologies such as HoloLens. Instead of the healthy people, three out of five had already used HoloLens; they are all between ages 20 and 35, including one female.

In the first step, the therapist was trained to set up the table with virtual objects, start the test, and decide whether or not to display some of the available parameters in real-time. The therapist can select six standard table configurations from a smartphone, as shown in Figure 14, to give more standardization to the data collected during testing.

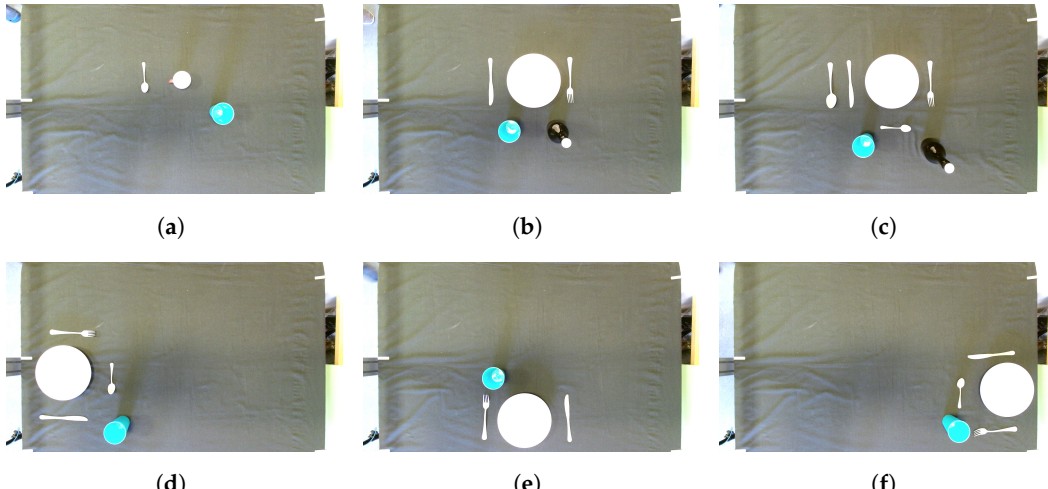

**Figure 14.** Different table setting configurations: (**a**–**c**) from a simple set-up to a complex one in the center of the table, and (**d**–**f**) from different angles.

The testing protocol was organized as follows:

(1). After completing a consent form, the tester receives an initial explanation of the task.
(2). Before starting, basal values of physiological data, such as heartbeat and breath rate, were estimated by acquiring data for 5 min.
(3). The therapist starts the tests in sequence: each tester must set the table in any configuration provided by the therapist.

All testers repeated the protocol for three consecutive days. Given the familiarity with the standard table-setting task and the ease of superimposing the real objects with the virtual ones, no initial training was necessary for the tester. We collected all the data acquired on different days in two populations: the one defined by healthy users and the one described by patients. The two-sample $t$-test [31] is used to compare whether the average difference for each selected parameter between the two populations is significant or if it is due to random effects. Before the $t$-test, we applied an initial variance test to check whether the two data samples were from populations with equal variances. In case of a negative outcome, it is replaced with Welch's formulas. The results accept the null hypothesis at the 5% significance only for breath rate and heart beat parameters. It means that there is no significant difference between patients and healthy testers for these two parameters. It can be attributed to the simplicity of testers' tasks and the test's short duration. In fact, it goes from an average duration value for healthy testers of 27 s to one of 59 s for patients. The difference in the other parameters allows the two populations to be distinguished. Figure 15 shows the boxcharts of errors in object placement and execution time.

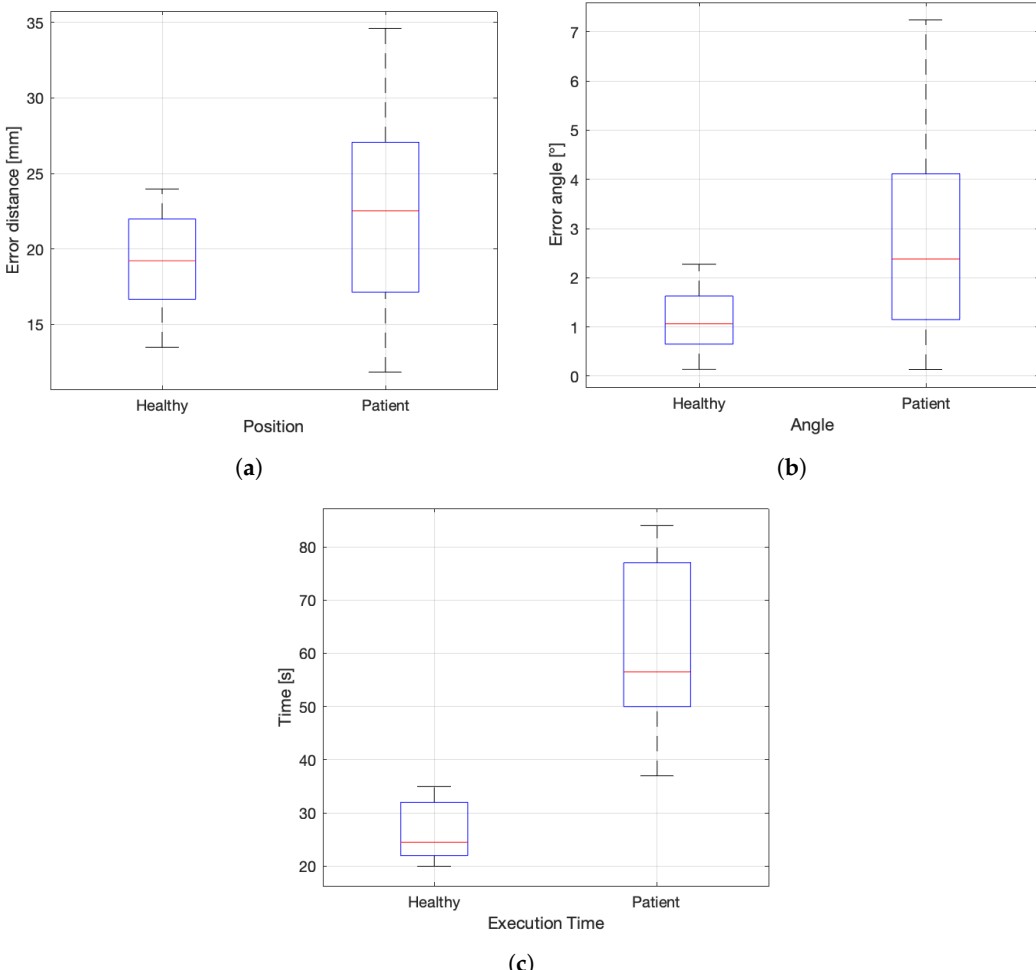

**Figure 15.** Boxcharts of the median error in (**a**) position (*p*-value = 0.001), (**b**) angle (*p*-value = 2.4 × 10$^{-7}$) and (**c**) execution time (*p*-value = 4.8 × 10$^{-5}$).

The difference in mean execution time between healthy testers and patients shown in Figure 15c is more significant than that related to errors in object placement (Figure 15a,b). In many human-performed tasks, the more precisely the assignment is to be accomplished, the slower it is. Fitts' law [32] reveals the correlation between speed and accuracy regarding human muscle movement. In our case, unlike a healthy person, for whom good results can be obtained in less time, i.e., with more speed, for a patient, even with more time, acceptable results can be obtained in terms of errors in object placement. No time limits were imposed during the test, but therapists only told patients to place the objects in the correct position. The above follows Fitts' law trade-off between speed and accuracy: to try to keep accuracy low, the maximum speeds and, therefore, the execution times between patients and healthy testers change. Figure 16 shows an example of the speed results at 6 Hz of setting the table in the configuration of Figure 14d.

Longer execution times for patients result in lower maximum speed. In fact, for the healthy tester in the example, the maximum speed is higher, and five-speed abrupt changes can be identified, each corresponding to the five objects in the selected configuration. As each object is grabbed from shelves, the speed remains high for the healthy subject, almost without slowing down during the grab control phase. The time to place the object in the correct position is also low and corresponds to the low-speed moments. For the patients, on the other hand, there are many more and smoother variations at low speeds, indicating continuous grabbing and releasing of objects without clean manipulation during the control phase in the final positioning of objects and grabbing them from the shelves.

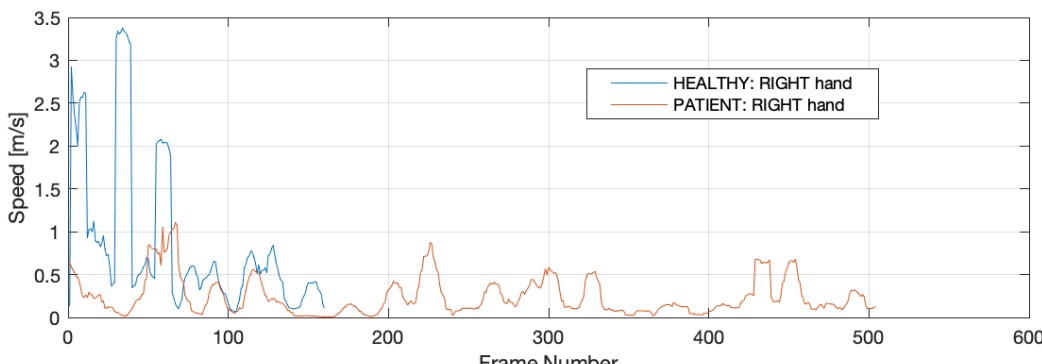

**Figure 16.** Example of speed comparison on the same test between a patient and a healthy tester.

For the same speed example, we show the result of the pressure distribution, as shown in Figure 17. The healthy tester usually puts all his weight on the leg on the side where he extends the arm he is using. On the other hand, for patients with stability problems, this is not true. The trend of the healthy subject, especially when he sets the table toward the lateral sides, follows a bimodal trend that can be identified with Warren Sarle's bimodality coefficient. Applying the BC to the data in Figure 17, we obtain the result in Figure 18 where, as might be expected, the BC is greater for the healthy tester for both feet.

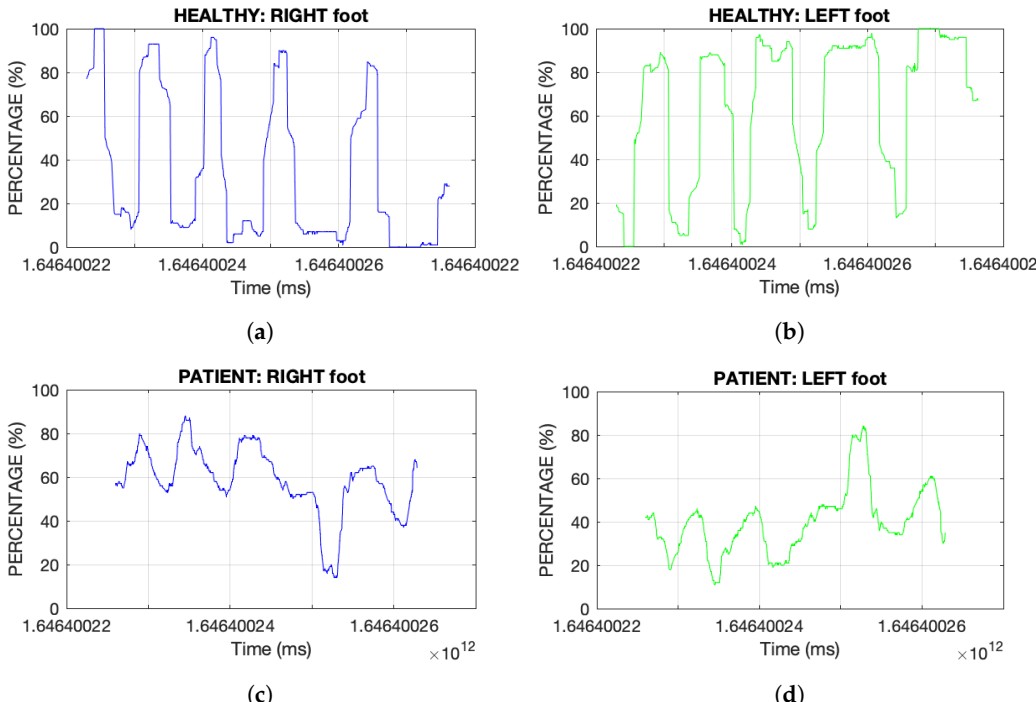

**Figure 17.** Example of pressure distribution on the same test between (**a**,**b**) a healthy tester and (**c**,**d**) a patient with right and left foot, respectively.

In addition, this testing campaign also defined the acceptability threshold of each parameter for patients. We used the results of healthy testers as acceptability thresholds, so, for example, an error of 18 mm for object position (Figure 15a) and 1° for its angle (Figure 15b) resulted acceptable for patients. Errors may be due to how the HMD glasses were worn or how the virtual images were displayed in AR.

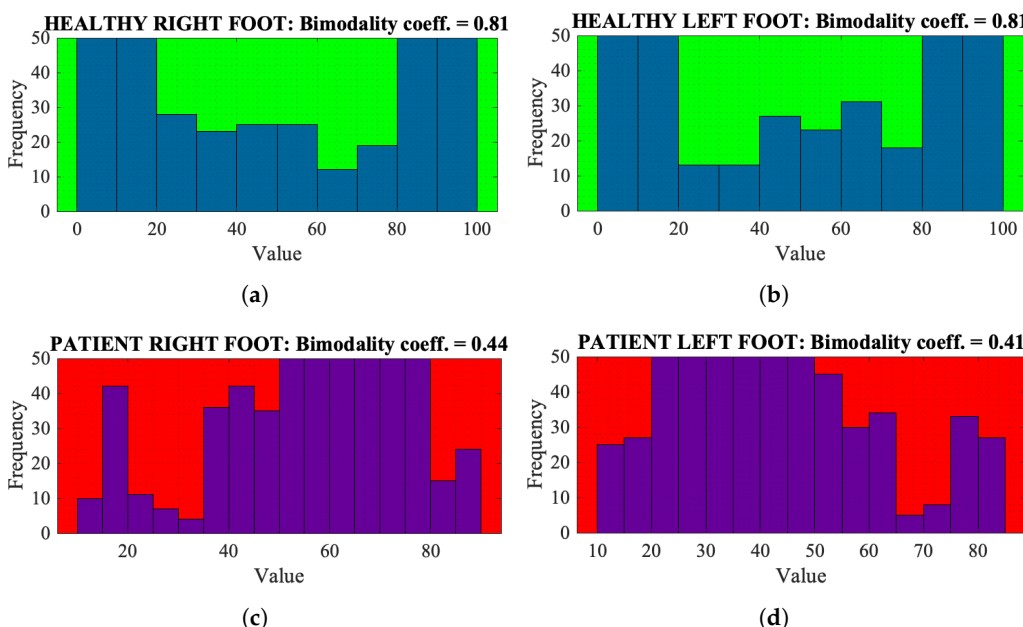

**Figure 18.** Example of Warren Sarle's bimodality coefficient of the same (**a**,**b**) healthy tester and (**c**,**d**) patient from Figure 17 data.

## 8. Offline Interface

One of the advantages of the designed AR framework is the possibility for therapists to have additional information available to assess patients in real-time and in the correct location near patients. However, they can save all the data collected during testing for further analysis. An interface was designed in MATLAB to read and visualize this information collected once the patient's name, day, and test number were selected. It allows therapists to have an overview of the entire test performed by patients, with the possibility of analyzing multiple parameters synchronized with each other, stopping or moving in time at will. In addition, offline analysis allows patients' performance to be compared even between tests performed at a distance of time. An example of how the offline post-processing interface for each tester looks is shown in Figure 19.

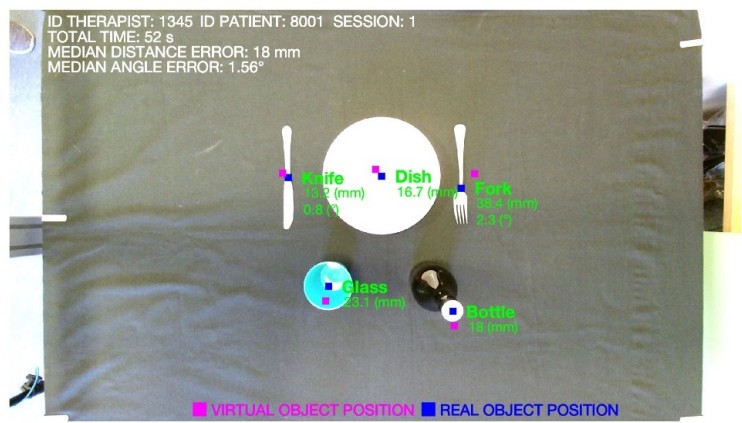

(**a**)

**Figure 19.** *Cont.*

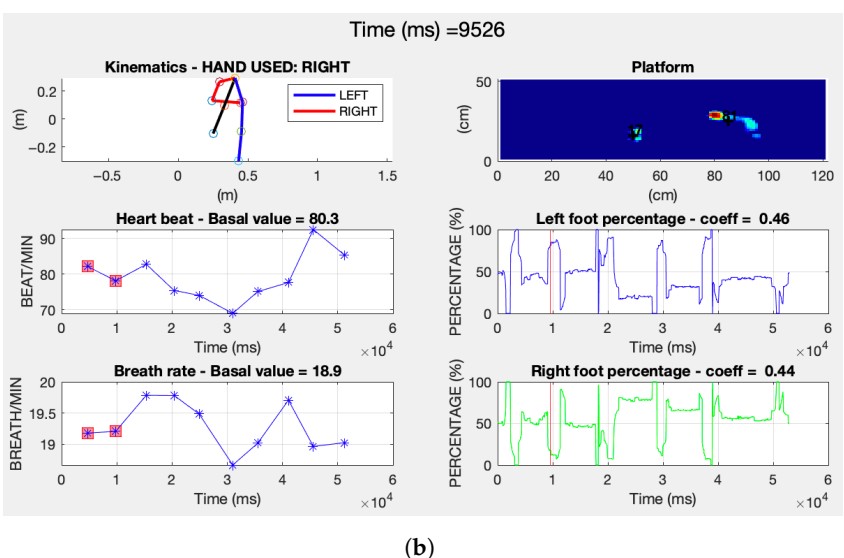

(**b**)

**Figure 19.** (**a**) Image with therapist and tester data, all errors in object placement and time of execution; (**b**) all other tester parameters are summarized in this second panel.

## 9. Conclusions

In this paper, an innovative AR multidimensional framework was developed in a shared AR environment where both therapist and patient have access to the same space-aligned VR. The therapist enhances the assessment of a patient's daily life activity, and through their interaction, it was possible to increase patient engagement and therapist involvement. The co-design of the prototype for the specific ADL of setting the table was realized in collaboration with clinical experts of the Villa Rosa Rehabilitation Hospital in Pergine Valsugana (Italy).

The calibrated setup ensures an uncertainty in object localization of 5 mm with a confidence level of 95% taking into account the higher eigenvalue multiplied by the factor $k = 2.4478$ [28] computed for two-dimensional covariance matrix and residual values due to estimated object rotations of less than 1°. We evaluated the designed framework using a user study with patients and healthy testers. It allowed the selection of significant parameters and their acceptance thresholds, as well as the goodness of the proposed method. The proposed framework was developed for the specific ADL of setting the table. However, it can also be applied in other AAL scenarios for metrological assessment of impaired or frail users and optimize the living environment itself.

It can be applied in OT to objectively evaluate treatment/training effectiveness in the clinical setting. In a future test campaign, we plan to use the prototype on other patients in parallel with their treatment and training to restore their autonomy with proper evaluation in the AUSILIA infrastructure.

**Author Contributions:** Conceptualization, M.D.C., A.L., I.B.III, F.P., G.M.A.G., J.B., M.M. and K.H.; Methodology, M.D.C., A.L., I.B.III and K.H.; Software, A.L. and I.B.III; Validation, M.D.C., A.L., I.B.III and F.P.; Formal analysis, M.D.C., A.L. and M.M.; Investigation, M.D.C., A.L., I.B.III and F.P.; Data curation, M.D.C., G.M.A.G. and J.B.; Writing—original draft, M.D.C., A.L. and I.B.III; Writing—review & editing, F.P., G.M.A.G., J.B., M.M. and K.H.; Visualization, M.D.C. and A.L.; Supervision, M.D.C., G.M.A.G., J.B. and K.H.; Project administration, M.D.C., G.M.A.G., J.B. and K.H. All authors have read and agreed to the published version of the manuscript.

**Funding:** This research received no external funding.

**Data Availability Statement:** The data presented in this study are available on request from the corresponding author. The data are not publicly available due to privacy or ethical restrictions.

**Conflicts of Interest:** The authors declare no conflict of interest.

**Ethical Statement:** All subjects gave their informed consent for inclusion before they participated in the study. The study was conducted in accordance with the Declaration of Helsinki, and the protocol was approved by the Ethics Committee of University of Trento (Project identification code: 2021-013).

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
