# Peer review of "Sharing Augmented Reality between a Patient and a Clinician for Assessment and Rehabilitation in Daily Living Activities†"

_information, doi:10.3390/info14040204_

Round 1

Reviewer 1 Report

1. The abstract should be rewritten and improved. The main problems existing in the current research of AR application in rehabilitation scenarios should be described in combination with the methods, main results and prospective application of the proposed AR framework.

2. The focus of the title and abstract description is not uniform. The title focuses on multidimensional evaluation, while the abstract focuses on an innovative Augmented Reality (AR) framework. Please unify the focus of the title and abstract description.

3. Please add the type, manufacturer and city of all equipment used in the article.

4. Please delete keywords that are not highly relevant to the main innovation of the article, such as user study, t-tests, etc.

5. Please unify the expression order and format of full names and abbreviations when proper nouns first appear.

6. In section 3.3, we suggest you to add the flow chart of data processing so that readers can better understand the data processing process in the experiment.

7. Figure 2 is the data transmission pipeline, not the data processing pipeline. We suggest you to modify the name and put it near the content of data transmission.

8. In section 5.1, we suggest you to add the flow chart of algorithm processing data so that readers can better understand the algorithm processing process.

9. In lines 259,346,356,362, please pay attention to the indentation before the paragraph.

10. The legend of figure 10b exceeds the icon range, please modify it.

Reviewer 2 Report

The authors present a AR framework for rehabilitation and skill assessment in ocupational therapy. Through the use of Hololens and Kinect devices, the therapist can interact/observe the patient/user and obtain useful information while the task is being performed. After the task completion, it is also possible to verify the patient/user performed based on a matlab tool also developed by the authors. The work is very interesting and the proposed platform is very useful. The paper is well writen and complete. In the next paragraphs, I suggest some points in order to improve even more the paper quality and readers' understanding.

Why do the authors use a time of flight camera just for capturing RGB full hd image (camera above the table)?

Please provide more detail (or reference) about how to connect two kinect v2 to a single PC at the same time and use them in real time, since this is a non trivial task.

"The first step is determining whether the areas between INIM and REFIM are similar within 30%." -> what happens if the patient has to manipulate an object that is outside the plane of the table? is this supported by the system? Do the object recognition runs in real time or only when the therapist presses the button to assess the error between virtual and real objects?

What happens if the patient places the object on the table in a different position than expected? for instance, if instead the bottle is placed in vertical position, it is placed horizontally? does the detection fail because the horizontal bottle cannot be matched to the objects in the database?

"Because of the familiarity with the task, no initial training was necessary for the tester." -> How is this task performed without the system? How does the therapist indicate the position of the objects to be placed without the system?

Please provide more detail regarding the duration of the tests. After the 5 minute warmup, how long did it take for users and patients to complete the tasks? Did this time diminished in the next days of testing?

"However, they can save all the data collected during testing for further analysis. " -> what is the influence of users' hands in the recognition of the objects? what about occlusion?

Since the therapist may use the offline matlab interface to visualize the complete task being performed after it was concluded, what was the real gain in having the therapist being able to follow the user performing the test in real time? Did it interfere during the test or it was basically an observational role? It seems that if the purpose is to scale the proposed test platform to other locations, a single Microsoft Hololens could be used instead of two. I believe this is also true for the kinect devices. Please ellaborate on this.

Please make clear that the current project passed an ethics committe board and that all tested users had to sign a consent form to participate on the tests.

More general comments and a few minor errors found are listed as follows.

"time, Fig. 5b." -> "time, shown in Fig. 5b."

"parameters Fig. 6." -> "parameters (Fig. 6)."

" set Fig. 7a," -> " set (Fig. 7a),"

"image Fig. 7b." -> "image (Fig. 7b)."

", Fig. 7c." -> ", as shown in Fig. 7c."

"traced [23], Fig. 7d" -> "traced [23], as shown in Fig. 7d."

"For this value to fall within an acceptable order of magnitude, both source images were cropped before this comparison with dimensions 50 % larger than the largest dimension between INIM and REFIM. " -> I did not understand this part. Could you improve the explanation please?

"Nevertheless, it has not been necessary to calibrate the camera at different heights because knowing the exact heights of the objects and the camera, with respect to the plane using trigonometric operations, we have always referred to the plane of the table. " -> this is very clever

" testers, Fig. 11." -> " testers, as shown in Fig. 11."

"cerebral ataxia," -> I believe the correct term is "cerebellar ataxia"

"Guillarm Barré" -> "Guillain Barré"

"smartphone, Fig. 12," -> "smartphone, as shown in Fig. 12,"

Is light blue really the color of the object in figure 12 or is it being highlighted? If the last is the case, why is it being highlighted?

"for parameters breath rate and heart beat." -> "for breath rate and heart beat parameters."

" placement Fig. 13a and Fig. 13b." -> " placement (Fig. 13a and Fig. 13b)."

"testers changes." -> "testers change."

"distribution, Fig. 15. " -> "distribution, as shown in Fig. 15. "

" for patients, with" ->  "for patients with"

"It allows the" -> "It allowed the"

"In the clinical setting," -> please remove this

Round 2

Reviewer 1 Report

After the revision, the content and logic of the article are presented better. But there are still a few small issues that need to be improved.

1. The writing format is easily confusing. In particular, the six items in "4. Specific ADL in the kitchen environment" still use the format "1. XXX". For example, Line226 is recommended to be modified to "(1). XXX".

The format of the annotation needs to be unified. For example, in Figure 14, (a) - (f) are marked, but they are not specified in the annotation. 
